# Identification of Candidate Genes for Meat Color of Chicken by Combing Selection Signature Analyses and Differentially Expressed Genes

**DOI:** 10.3390/genes13020307

**Published:** 2022-02-07

**Authors:** Jiahong Sun, Xiaodong Tan, Xinting Yang, Lu Bai, Fuli Kong, Guiping Zhao, Jie Wen, Ranran Liu

**Affiliations:** State Key Laboratory of Animal Nutrition, Institute of Animal Sciences, Chinese Academy of Agricultural Sciences, Beijing 100193, China; sunjiahong416@163.com (J.S.); tanxiaodong08@163.com (X.T.); yangxinting0704@163.com (X.Y.); blu1130@163.com (L.B.); kongfuli617@163.com (F.K.); zhaoguiping@caas.cn (G.Z.); wenjie@caas.cn (J.W.)

**Keywords:** breast muscle, glycolysis, heme pigments, lipid metabolism

## Abstract

Meat color, an important index of chicken quality, is highly related to heme pigment, glycolysis, and intramuscular fat metabolisms. The objective of this study is to obtain candidate genes associated with meat color in chickens based on the comparison of fast-growing, white-feathered chickens (Line B) and slow-growing, yellow-feathered chickens (Jingxing Yellow), which have significant differences in meat color. The differentially expressed genes (DEGs) between Line B and Jingxing Yellow were identified in beast muscle. The fixation index (*F_ST_*) method was used to detect signatures of positive selection between the two breeds. Screening of 1109 genes by the *F_ST_* and 1317 candidate DEGs identified by RNA-seq. After gene ontology analysis along with the Kyoto Encyclopedia of Genes and Genomes, 16 genes associated with glycolysis, fatty acid metabolism, protein metabolism, and heme content were identified as candidate genes that regulate the color of chicken breast meat, especially *TBXAS1* (redness), *GDPD5* (yellowness), *SLC2A6* (lightness), and *MMP27* (lightness). These findings should be helpful for further elucidating the molecular mechanisms and developing molecular markers to facilitate the selection of chicken meat color.

## 1. Introduction

The quality of chicken meat is affected by many factors, especially color, intramuscular fat (IMF) content, juiciness, and tenderness [1,2,3,4]. Meat color is not only an important index of the appearance of chicken but also an external manifestation of physiological, biochemical, and microbiological changes to muscle tissues. Chickens can be broadly classified as fast-growing, white-feathered, and slow-growing color-feathered varieties. In China, slow-growing yellow-feathered chickens are more popular because of their better flavor and higher IMF content [5]. However, factors underling differences in meat color among different chicken breeds remain unclear.

Meat color is mainly determined by pigment content and is measured visually by the relative degrees of redness, yellowness, and lightness. Carotenoid metabolism is considered critical to the yellowness of chicken meat [6,7]. Myoglobin is the primary pigment responsible for the redness of meat, as the basic molecular structure is composed of heme [8], while the redness and lightness qualities are related to muscle structure. Postmortem anaerobic glycolysis leads to massive acid accumulation, which results in protein denaturation and changes to the physical appearance of meat [9]. Spatial changes to the structure of muscle tissues lead to decreased muscle oxygen content and changes to the reflection of light, thereby affecting perceptions of redness and lightness [10]. The lightness and yellowness qualities of chicken meat are related to the amount of lipids in the muscle tissues. Chartrin et al. [11] found that duck breast muscle containing high lipid levels was paler with greater intensity in yellowness. Collectively, the findings of these studies indicate that the color of chicken meat is associated with pigment content, post-slaughter glycometabolism, lipid metabolism, etc.

Recent progress in chicken genomics has led to the identification of genes and mutations underlying phenotypic variation of complex traits. Fine-mapping of quantitative trait loci and gene expression analysis conducted by Taniguchi et al. [12] suggested that *NUDT7* is the most likely candidate gene responsible for the redness of pork. In addition, overexpression of *NUDT7* was reported to downregulate heme biosynthesis in rat L6 myoblasts. Le Bihan-Duval et al. [13] revealed that *BCMO1* is associated with the yellowness of breast muscle, mainly by altering the contents of the yellow pigments lutein and zeaxanthin. As with other complex traits, meat color is regulated by the interactions of various and often unknown genes, thus warranting further investigations.

In the present study, two chicken breed/lines with significantly different meat color traits (i.e., fast-growing, white-feathered, and slow-growing yellow-feathered varieties) were used to identify candidate differentially expressed genes (DEGs), by RNA sequencing (RNA-seq) and positively selected genes (PSGs), based on the fixation index (FST) of genome sequencings. Candidate genes for meat color were screened out within the overlap genes between PSGs and DEGs. Those candidate genes were selected related to pigment content, glycometabolism after slaughter, and lipid metabolism. This study aims at providing a theoretical basis for understanding the genetic mechanism of chicken meat color and the development of molecular markers to facilitate the selection of chicken meat color.

## 2. Materials and Methods

### 2.1. Ethics Statement

The study protocol was approved by the Ethics Review Committee of the Institute of Animal Sciences (IAS) of the Chinese Academy of Agricultural Sciences (CAAS) (reference no. IAS2019-44) and conducted in strict accordance with the Regulations for the Administration of Affairs Concerning Experimental Animals established by the Chinese Ministry of Science and Technology.

### 2.2. Animals

Chickens from two fast-growing, white-feathered chickens (Line B and Arbor Acres) and two slow-growing yellow-feathered chickens (Jingxing Yellow and Beijing-you) were used in this study. Line B chickens, a fast-growing, white-feathered pure line, were produced and raised by Foshan Gaoming Xinguang Agricultural and Animal Industrials Co., Ltd. (Foshan, China) [14]. Jingxing Yellow chickens, a typical slow-growing yellow-feathered line, were obtained from the IAS (CAAS) [15]. The basal diet was formulated based on the National Research Council (1994) requirements and the Feeding Standards of Chickens established by the Ministry of Agriculture, Beijing, China (2004). Feed and water were provided ad libitum throughout the experiment period. The meat quality traits of Line B and Jingxing Yellow chickens (*n* = 100 each) were measured at market age (42 and 98 days, respectively). RNA and genome sequencing of breast meat samples of the two breeds was conducted as described in Section 2.4 and Section 2.5, respectively.

Arbor Acres broilers, commercial fast-growing, white-feathered chickens, were obtained from Yijia Agricultural Development Co., Ltd. (Chengde, China). Beijing-you chickens, a typical slow-growing yellow-feathered line, were obtained from the IAS (CAAS). The Arbor Acres and Beijing-you chickens were both raised under the same environmental conditions [16] by the Changping Experimental Base (IAS, CAAS). The basal diet was formulated based on the National Research Council (1994) requirements and the Feeding Standards of Chickens established by the Ministry of Agriculture, Beijing, China (2004). Feed and water were provided ad libitum throughout the experiment period. Meat quality traits of 27 Arbor Acres and 27 Beijing-you chicken were measured at 42 days, and samples were collected for reverse transcription quantitative real-time polymerase chain reaction (RT-qPCR) analysis of candidate genes associated with the color of breast meat as described in Section 2.8.

### 2.3. Meat Color Measurements

The meat quality indices of meat color and pH were measured at 15 min and 24 h post slaughter, respectively. The meat color (L*, a*, b*) of breast muscle was measured using the CIELAB color space system [17], which consists of opponent-color scales based on the opponent-color theory of human color vision, where a* (redness) indicates redness when positive and greenness when negative, b* (yellowness) indicates yellowness when positive or blueness when negative, and L* (lightness) describes the relationship between reflected and absorbed light, with a value of 100 for white and 0 for black. The pH value of breast muscle was measured using a pH meter (HI8424; Hanna Instruments, Woonsocket, RI, USA) [18].

### 2.4. RNA-Seq

Six breast muscle samples of Line B chickens and 10 of Jingxing Yellow chickens were randomly selected for mRNA sequencing, which was conducted by Annoroad Gene Technology Co., Ltd. (Beijing, China). In brief, mRNA was enriched by binding the mRNA poly-A tail to oligo(dT)-coated magnetic beads and fragmenting into small pieces. Single- and double-stranded cDNA were synthesized using mRNA as a template. The double-stranded cDNA was purified using the QIAQuick PCR purification kit (QIAGEN, Inc., Valencia, CA, USA). After purification, end repair, and ligation to sequencing adapters, agarose gel electrophoresis was used for fragment size selection. Then, PCR enrichment was performed to obtain the final cDNA library. RNA-seq was performed with the Illumina NovaSeq 6000 System equipped with an S2 flow cell (Illumina, Inc., San Diego, CA, USA), in accordance with the manufacturer’s instructions, and 150-bp paired-end reads were generated. Sequence adapters and low-quality reads (read quality < 30) were removed with the Trimmomatic (v0.32) trimming tool. Quality control of the raw sequencing data was conducted with the FastQC (v 0.11.9) quality control tool for high-throughput sequence data. The acquired reads were mapped to the chicken reference genome GRCg6a (GCA_000002315.5) using HISAT2 software (v 2.1.0), and transcripts were assembled with String Tie software (v 1.3.2d). To quantify the expression of each gene, alignment results were analyzed with HTSeq (v 0.6.0) high-throughput sequence analysis software. Analysis of differential expression of transcripts was performed with the DESeq2 package (v 1.24.0). Genes with a probability (*p*) value <  0.05 and |fold change|  ≥ 2 were considered DEGs.

The RNA-seq data of the breast muscle of Jingxing Yellow chickens are included in a previous report by our group [15] and are available at https://bigd.big.ac.cn/gsa/, accessed on 10 July, 2021 (accession data code CRA001908). The RNA-seq data of the breast muscle of Line B chickens are available at https://bigd.big.ac.cn/gsa/, accessed on 10 July, 2021 (accession data code CRA005468).

### 2.5. Genome Sequencing and Selection Signature Analysis

The whole-genome sequencing data of 520 Jingxing Yellow chickens were obtained from a previous study by Liu et al. [19] and are available at https://bigd.big.ac.cn/gsa/, accessed on 10 July, 2021 (accession data codes CRA002643 and CRA002650). The whole-genome sequencing data of 230 broilers were obtained from a previous study by Li et al. [14] and are available at https://bigd.big.ac.cn/gsa/, accessed on 10 July 2021 (accession data code CRA002454).

Then, 150-bp paired-end reads obtained from all broilers were re-sequenced with the Illumina NovaSeq 6000 platform by Zhejiang Annoroad Biotechnology Co., Ltd. (Hangzhou, China). Clean sequencing data were aligned to the chicken reference genome GRCg6a (GCA_000002315.5) with the Burrows–Wheeler Aligner-MEM sequence mapping tool [20]. PCR duplicates were removed. Finally, single-nucleotide polymorphisms (SNPs) were identified with the Genome Analysis Toolkit (v3.5) [21]. The filtering settings were as follows: variant confidence score < 30.0, QualByDepth < 2.0, ReadPosRankSum < −8.0, total depth of coverage < 4.0, and FisherStrand > 60.0. The SNP data were controlled using PLINK software (v 1.90beta) with the following parameters: geno, 0.1; hwe, 1 × 10^−6^; maf, 0.01; mind, 0.1; and chr, 1–28. The *F_ST_* index was calculated with the VCFtools package (v 0.1.13) [22] with a window size of 40 K and window-step of 10 K. Windows with the top 5% *F_ST_* values of each chromosome were considered as candidate regions of selection signatures.

### 2.6. Pathway Analysis

Kyoto Encyclopedia of Genes and Genomes (KEGG) and gene ontology (GO) enrichment analyses were performed with the KEGG Orthology-Based Annotation System 3.0 [23] to identify enriched pathways of the DEGs and PSGs. A *p*-value of 0.05 was set as the threshold for significant enrichment of DEGs.

### 2.7. RT-qPCR Validation of Candidate Genes

The 16 candidate genes were verified by RT-qPCR with breast meat samples in the four breeds/lines described in Section 2.2. The number for RT-qPCR of Jingxing Yellow and Line B is both 12, 14 of Beijing-you chickens and 13 for Abor Acres. Briefly, 1.8 μg of total RNA of each sample was reverse-transcribed into cDNA for RT-qPCR analysis with specific primers (Table 1) from retrieved sequences at or just outside exon/exon junctions to avoid the amplification of residual genomic DNA using the Primer-BLAST tool (https://www.ncbi.nlm.nih.gov/tools/primer-blast/, accessed on 15 July, 2021. Verified primers for all candidate genes are shown in Appendix A. The expression levels of the target genes were normalized to RPL32. The RT-qPCR amplifications were conducted in triplicate with the QuantStudio™ 7 Flex Real-Time PCR System (Applied Biosystems, Carlsbad, CA, USA) and an amplification protocol that consisted of an initial denaturation step at 95 °C for 3 min, followed by 40 cycles at 95 °C for 3 s and 60 °C for 34 s. The results were analyzed with the 2^−∆∆CT^ method [24].

### 2.8. Statistical Analyses

The results are presented as the mean ± standard error of the mean. Differences between and among groups were assessed using the Student’s *t*-test and analysis of variance, respectively. All statistical analyses were conducted using IBM SPSS Statistics for Windows, version 26.0. (IBM Corporation, Armonk, NY, USA). A *p*-value < 0.05 was considered statistically significant.

## 3. Results

### 3.1. Differences in Meat Quality between Jingxing Yellow and Line B Chickens

Differences in the color of breast meat between Jingxing Yellow and Line B chickens are summarized in Table 2. As compared with Jingxing Yellow chickens, redness was significantly higher, and yellowness was significantly lower in Line B chickens. Lightness was significantly reduced in Line B chickens at 15 min post slaughter and relatively increased at 24 h post slaughter. Additionally, the pH of Line B chickens was significantly higher than that of Jingxing Yellow chickens.

To testify phenotype divergency and candidate genes related to meat color, two breeds of chickens were raised under the same environmental conditions. As shown in Table 2, as compared with the Arbor Acres, lightness was significantly enhanced at 15 min post slaughter and significantly decreased at 24 h post slaughter. The yellowness of breast muscle of the Beijing-you chickens was significantly higher at 42 days. Additionally, the pH at 24 h post slaughter was significantly lower for Beijing-you chickens as compared with Arbor Acres. Both two comparisons show consistent patterns.

### 3.2. Screening of DEGs Based on RNA Sequencing Data

In total, 1317 DEGs (Appendix A) were identified between the breast meat samples from Line B and Jingxing Yellow chickens. The GO and KEGG enrichments of these genes are summarized in Figure 1. The enriched genes were mainly associated with fatty acid metabolism (extracellular matrix (ECM) receptor interaction pathway; focal adhesion; Wnt signaling pathway), amino acid metabolism (proteolysis; glycine, serine, and threonine metabolism; alanine, aspartate, and glutamate metabolism), and binding to heme and iron (heme binding; iron ion binding).

### 3.3. Genes Associated with Selection Signatures

In total, 1109 genes were identified as PSGs (Appendix A). PSGs were enriched in terms associated with fatty acid metabolism (fatty acid biosynthesis; Wnt signaling pathway; focal adhesion), protein turnover (mTOR signaling pathway; proteolysis), and binding to heme and iron (heme binding; iron ion binding) (Figure 2).

### 3.4. Candidate Genes for Meat Color of Breast Muscle in Chickens

In total, 16 candidate genes, identified by overlapping of DEGs and PSGs, were involved in pathways related to glycolysis, fatty acid metabolism, protein metabolism, and binding to heme and iron (Table 3).

The significant differences in the expression patterns of the 16 overlapping genes were validated in the four breeds/lines. Between Line B and Jingxing Yellow chickens, the differential expression trends of eight genes (*GDPD5*, *MMP27*, *SLC2A6, TBXAS1 ADAMTSL2*, *FGF2*, *NFATC2,* and *PAPPA2*) are basically consistent with that obtained by transcriptome sequencing (Figure 3, Appendix A). As shown in Figure 4, there were significant differences (*p* < 0.05) in the expression patterns of *GDPD5*, *MMP27*, *SLC2A6*, and *TBXAS1* between Arbor Acres and Beijing-you chickens as well.

## 4. Discussion

There are significant differences in meat quality indices between fast-growing, white-feathered and slow-growing yellow-feathered chickens, especially sensory and nutritional components [25]. Line B chickens used in the current study is a fast-growing, white-feathered pure line. Arbor Acres used are the commercial broilers characterized by fast growing as well. Beijing-you chickens and Jingxing Yellow are both typical slow-growing yellow-feathered chickens with suitable meat quality. They are representative ones of fast-growing chickens and slow-growing chickens. Differences in the color of chicken meat are driven by a series of factors, including age, breed, rearing conditions, diet, and processing parameters [26]. The color of chicken breast meat seems to be independent of sex [26]. Hence, the sex of the chickens used in this study was not considered. The breast meat of the slow-growing chickens was generally darker and more yellowish, while that of the fast-growing chicken was lighter with greater redness. These trends in meat color are generally the same regardless of whether there are differences in rearing conditions, use of the same breed, or measurements at the same age, indicating that in addition to the breed and breeding conditions, genetic factors play key regulatory roles in the determining the color of chicken breast meat. Selective sweep analysis was used to further screen selected regions of the genome and transcriptome data to identify candidate DEGs that regulate the color of chicken meat.

Meat color is regulated by complex interactions between biological traits and biochemical processes during the conversion of muscle to meat. KEGG and GO enrichment analyses identified 1317 DEGs and 1109 PSGs significantly enriched in various metabolic processes, signaling pathways, cellular processes, and glucose metabolism. High concentrations of myoglobin, the oxygen consumption rate, and oxygenation of myoglobin are associated with the redness of muscle tissues [10]. Among the significantly enriched GO terms, heme binding and iron binding (heme binding; iron ion binding) were especially enriched. Heme iron, which is mainly found in myoglobin in muscle tissue, contributes to the desirable bright red color of meat, but also brown, which is the most undesirable color [27]. *TBXAS1* was identified as a candidate gene associated with redness. *TBXAS1* encodes thromboxane synthase, an enzyme in the arachidonic acid cascade that produces thromboxane A2, which plays a key role in platelet aggregation [28]. A study of malignant mesothelioma conducted by Daisuke [29] found that *TBXAS1* gene expression was down-regulated in rats with iron-induced mesothelioma, and iron overload was associated with *TBXAS1* down-regulation.

Yellowness is related to the IMF content. Symeon et al. [30] reported greater IMF content and yellowness of the breast meat of the capon, a castrated or neutered rooster. Thomas et al. [31] found that selective breeding of lambs with low muscle density can increase the IMF content and improve meat color and other quality parameters and that the lipid deposition was associated with the yellowness of the meat. In the present study, lipid metabolism pathways were significantly enriched between the white-feathered lines and yellow-feathered Jingxing Yellow chickens.

Two KEGG pathways related to metabolism (ECM-receptor interaction, focal adhesion) were enriched. Focal adhesions and ECM-receptor interactions are associated with cell junctions, which adhere the cytoskeleton to the ECM [32], and are also related to lipid metabolism and IMF deposition [7]. Other DEGs have been associated with ECM-receptors related to omental and subcutaneous IMF deposition [33,34]. Yellowness is closely associated with lipid oxidation [35]. Among the DEGs, *GDPD5* encodes a glycerophosphodiester phosphodiesterase involved in glycerol metabolism and related to phosphoric diester hydrolase activity [36].

Lightness of meat is influenced by the protein composition of the surrounding medium (sarcoplasm and extracellular space), muscle structure, and cytoskeleton lysis, which also results in the redness of meat by influencing the ability of oxygen to diffuse into muscle tissues [10]. However, these parameters were also influenced by changes in pH, which resulted in glycolysis under postmortem hypoxia. *SLC2A6*, one of the DEGs related to glycolysis, which was also screened out by selection signature analyses, plays a key role in glucose transport. A recent study [37] reported that *SLC2A6* (*GLUT6*) was a modulator of glycolysis in inflammatory macrophages.

The reflection of meat is related to protein dissolution, the protein turnover rate, and amino acid metabolism [38]. The GO term proteolysis and the KEGG lysosome pathway related to protein solution were also enriched. *MMP27* encodes the protein matrix metallopeptidase 27, which is involved in the breakdown of the ECM, resulting in changes to the extracellular space that influence the lightness of meat color.

## 5. Conclusions

In this study, the yellowness phenotype was apparent in the breast meat of yellow-feathered chickens, while lightness, redness, and pH were comparatively enhanced in the breast meat of white-feathered chickens. Selection signature and RNA-seq analyses were performed to identify 16 candidate DEGs that regulate differences in the color of chicken breast meant. Some genes, such as *TBXAS1*, *GDPD5*, *SLC2A6*, and *MMP27,* can be regarded as important candidate genes for future studies. These results enhance our understanding of the role of natural selection in shaping genomic variation and genes contributing to differences in the chicken meat color.

## Figures and Tables

**Figure 1 genes-13-00307-f001:**
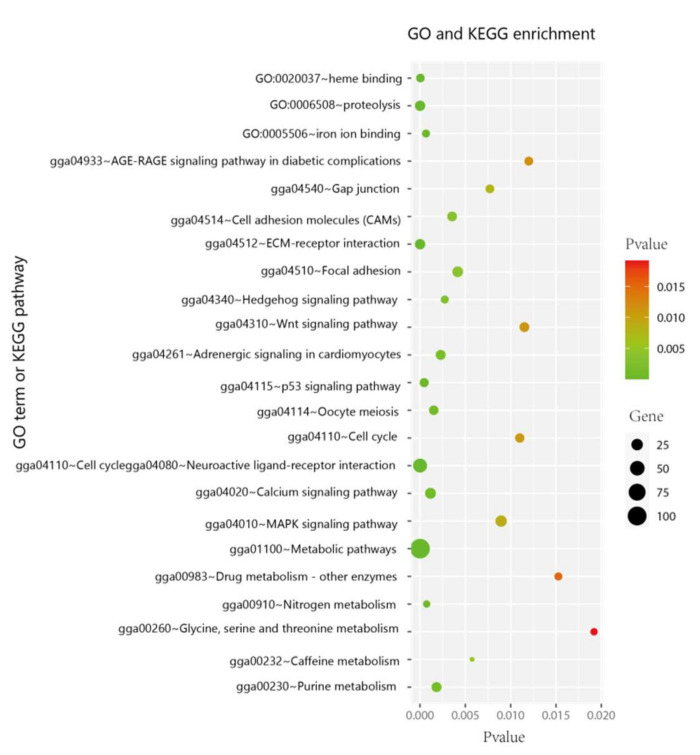
GO- and KEGG-enriched items associated with fatty acid metabolism, amino acid metabolism, and binding to heme and iron. The spot size represents the enriched gene number, and the spot color represents the *p*-value.

**Figure 2 genes-13-00307-f002:**
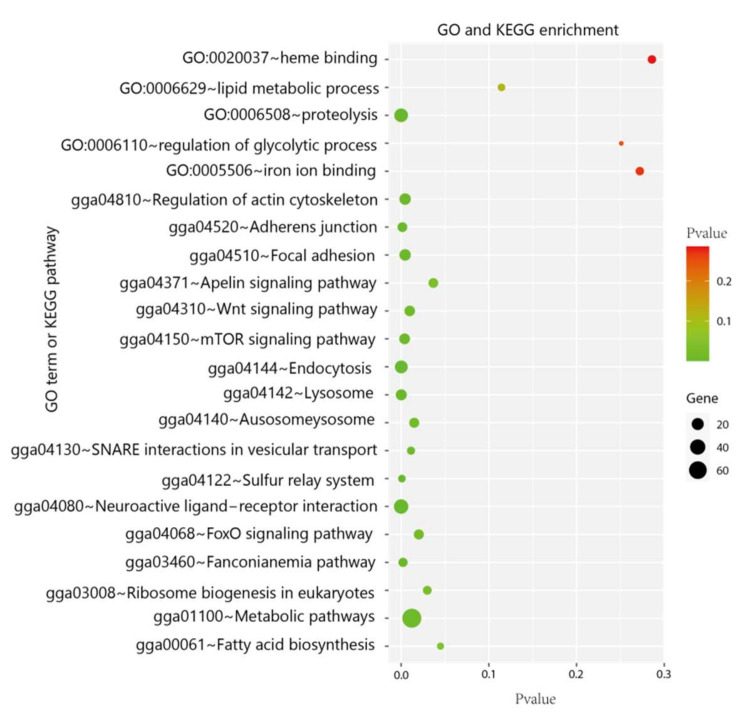
GO- and KEGG-enriched items associated with fatty acid metabolism, protein turnover, and binding to heme and iron. The spot size represents the enriched gene number, and the spot color represents the *p*-value.

**Figure 3 genes-13-00307-f003:**
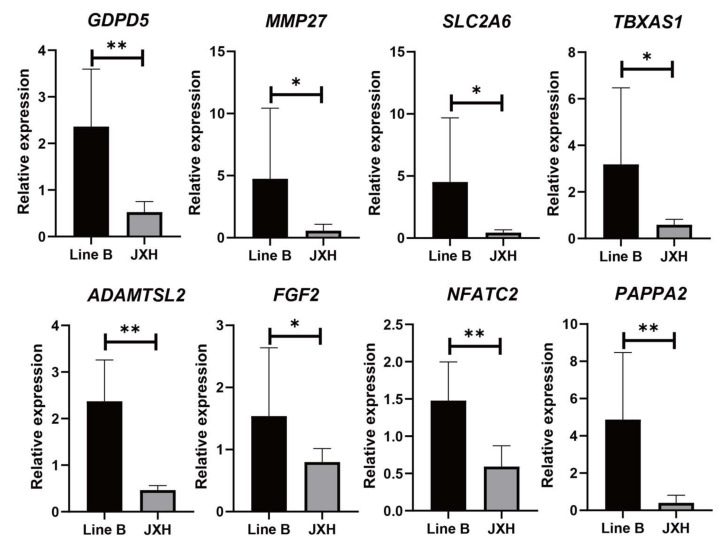
Verification of candidate gene expressions by RT-qPCR in Line B and Jingxing Yellow chickens (JXH), * *p* < 0.05, ** *p* < 0.01.

**Figure 4 genes-13-00307-f004:**
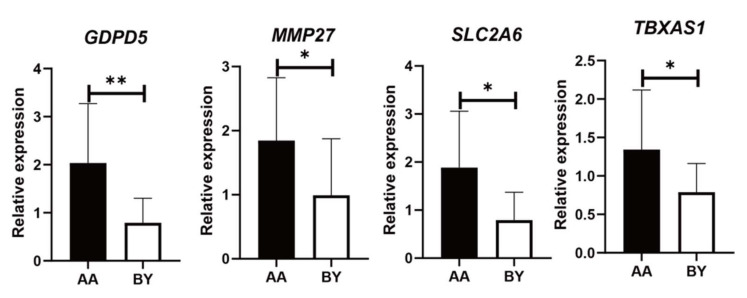
Verification of candidate gene expressions by RT-qPCR in Arbor Acres (AA) and Beijing-you chickens (BY), * *p* < 0.05, ** *p* < 0.01.

**Table 1 genes-13-00307-t001:** Genes and primers for RT-qPCR.

Gene Name	Forward Primers (5′-3′)	Reverse Primers (5′-3′)
SLC2A6	TTCCTTGGGGTTGTGGAGTT	AAGACATTCCCAGCGCAGAT
MMP27	CCAACCGTCCCTACATCACC	ACTCGCCACAAGTGTCTTCC
TBXAS1	GCTGTGCTGGGAGAAGATGT	CACTGTGCCAGCTTTCAGTG
GDPD5	TTTTTATATTCCAGAAGTGGCGCT	TCTTGACATCCCGACTGGAC
RPL32	AGTTCATCCGCCACCAGTCTGAT	GCTTCGTCTTCTTGTTGCTCCCATA

**Table 2 genes-13-00307-t002:** Differences in meat color indices between yellow-feathered and white-feathered chickens.

Parameter			15 min				24 h			
			L*	a*	b*	pH	L*	a*	b*	pH
Breed and age	98 days	Jingxing Yellow	55.28 ± 3.55 ^a^	8.57 ± 1.67 ^b^	14.53 ± 2.32 ^a^	5.55 ± 0.57 ^b^	54.09 ± 3.8 ^b^	7.39 ± 2.07 ^b^	15.51 ± 2.48 ^a^	5.68 ± 0.6 ^b^
	42 days	Line B	51.98 ± 1.99 ^b^	10.8 ± 1.17 ^a^	12.64 ± 1.95 ^b^	6.31 ± 0.2 ^a^	58.76 ± 3.11 ^a^	11.52 ± 1.32 ^a^	14.04 ± 1.88 ^b^	5.82 ± 0.19 ^a^
*p*-value			<0.0001	<0.0001	<0.0001	<0.0001	<0.0001	<0.0001	<0.0001	0.032
Breed and age	42 days	Beijing-you	53.51 ± 2.37 ^a^	11.79 ± 1.75	18.01 ± 2.05 ^a^	5.75 ± 0.2 ^b^	50.74 ± 9.30 ^b^	9.56 ± 1.92	16.98 ± 1.87 ^a^	5.88 ± 0.1 ^b^
	42 days	Arbor Acres	52.29 ± 2.93 ^b^	12.41 ± 14.06	11.65 ± 1.87 ^b^	6.00 ± 0.19 ^a^	55.03 ± 2.19 ^a^	10.65 ± 1.89	13.41 ± 1.7 ^b^	6.00 ± 0.14 ^a^
*p*-value			0.016	0.800	<0.0001	<0.0001	0.021	0.080	<0.0001	<0.0001

Data are expressed as the mean ± SEM. Different superscript letters within rows indicate significant difference.

**Table 3 genes-13-00307-t003:** The 16 candidate genes identified from associated pathways.

GO Entries/KEGG Pathways		Gene Name
GO:0006110	Regulation of glycolytic process	*SLC2A6* (Solute Carrier Family 2 Member 6)
gga04142	Lysosome	*ADAMTSL2* (ADAMTS-Like 2)
GO:0006508	Proteolysis	*TLL2* (Tolloid-Like Protein 2)
		*THSD4* (Thrombospondin Type 1 Domain Containing 4)
		*MMP7* (Matrix Metallopeptidase 7)
		*MMP27* (Matrix Metallopeptidase 27)
		*PHEX* (Phosphate-Regulating Endopeptidase Homolog X-Linked)
		*PAPPA2* (Pappalysin 2)
		*XPNPEP2* (X-Prolyl Aminopeptidase 2)
		*DNASE2B* (Deoxyribonuclease 2 Beta)
gga04150	mTOR signaling pathway	*COL1A2* (Collagen Type I Alpha 2 Chain)
GO:0005506	Iron ion binding	*TBXAS1* (Thromboxane A Synthase 1)
GO:0020037	Heme binding	*XDH* (Xanthine Dehydrogenase)
gga04310	Wnt signaling pathway	*NFATC2* (Nuclear Factor Of Activated T Cells 2)
		*DAAM2* (Disheveled-Associated Activator Of Morphogenesis 2)
		*MMP7* (Matrix Metallopeptidase 7)
gga04510	Focal adhesion	*FGF2* (Fibroblast Growth Factor 2)
GO:0006629	Lipid metabolic process	*GDPD5* (Glycerophosphodiester Phosphodiesterase Domain Containing 5)

## Data Availability

The raw whole-genome sequencing data and RNA-seq data reported in this article have been deposited in the Genome Sequence Archive in BIG Data Center (https://bigd.big.ac.cn/gsa, accessed on 10 July, 2021). The accession number of raw whole-genome sequencing data for Line B is CRA002454, for Jingxing Yellow chickens are CRA002643, CRA002650. The accession number of raw RNA-seq data for Line B chickens is CRA005468, for Jingxing Yellow chickens is CRA001908.

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
