# Peer review of "Identification of Candidate Genes for Meat Color of Chicken by Combing Selection Signature Analyses and Differentially Expressed Genes"

_genes, 2022, doi:10.3390/genes13020307_

Round 1
Reviewer 1 Report
-In the introduction and the aim of the research, no information has been provided on what effect the research results described in the paper will have on the quality of poultry meat? In section 2.7, RT-qPCR validation was planned, but there is no reference in the results.
Reviewer 2 Report
The Authors analyses two chicken breed/lines with significantly different meat color traits by RNA sequencing (RNA-seq) and positively selected genes (PSGs), to identify candidate differentially expressed genes (DEGs) for meat color.
Here are some suggestions that the Authors can consider improving the manuscript.
Introduction
Line 56: change “Elisabeth et al. [13]” with “Le Bihan-Duval et al. [13]”.
Materials and Methods
Please add a table with number of samples for the four commercial lines. Are animals of both sexes?
Line 74 to 95: please add the breeding and feeding condition of the two fast-growing white-feathered chickens (Line B and Arbor Acres) and the two slow-growing yellow-feathered chickens (Jingxing Yellow and Beijing-You).
Line 81: Line B and Jingxing Yellow chickens were measured for meat quality traits at age of 42 and 98 days, respectively. Why the meat quality traits of Arbor Acres and Beijing-You chicken were measured at 42 days only?
Line 107-: why the Authors focus on Line B and Jingxing Yellow and do not consider the other two lines for RNA-seq?
Results
Line 182- : I suggest to shows the results of meat quality in the four commercial line in a unique section and combine tables 2 and 4.
Similarly, in the section “Screening of DEGs based on RNA sequencing data” the results for the two fast-growing white-feathered chickens (Line B and Arbor Acres) and in the two slow-growing yellow-feathered chickens (Jingxing Yellow and Beijing-You) should be described together. I think the data should be added to evaluate the differences between slow-growing yellow-feathered and fast-growing white-feathered chickens.
The Authors presented at page 11, the significant differences in the expression patterns of the 16 overlapping genes validated by RT-qPCR, but they used data for Arbor Acres chickens and Beijing-You chickens. Why they do not validate the same genes on Line B and Jingxing Yellow? Again, I think the data of Line B and Jingxing Yellow lines should to be added to confront the differences in the expression patterns of the 16 genes on the four lines.
Discussion
In general, the “Discussion” is good, but I think is too general and not focalized on the four lines. The results obtained in the two fast-growing white-feathered chickens and in the two slow-growing yellow-feathered chickens are only marginally discussed.
Round 2
Reviewer 2 Report
The authors addressed most of the reviewer concern and made appropriate changes to the manuscript. I think the questions raised were now satisfactorily answered.
I only suggest adding the table of number and gender of each chickens in each part as supplementary material.